# Learning High-Order Substructure Association from Molecules with Transformers

## Abstract

Molecular graphs are commonly represented using SMILES (Simplified Molecular Input Line Entry System) strings, enabling the transformation of molecular graphs into token sequences. While transformers—powerful neural networks originally developed for natural language processing—have been adapted for learning molecular representations from SMILES by predicting masked tokens, they have yet to achieve competitive performance on ADMET benchmark datasets crucial for assessing drug properties such as absorption, distribution, metabolism, excretion, and toxicity. This paper identifies the challenge that traditional random token masking in SMILES overlooks essential molecular substructures, leading transformers to focus on superficial correlations between individual tokens rather than their relationships within substructures. We propose a novel approach that enhances transformers' capability to recognize molecular substructures by introducing a substructure-aware masking strategy alongside a new learning objective. This method embeds substructure information directly into the masking and prediction process, allowing the model to predict specific subgraphs instead of random tokens. Our experiments demonstrate that transformers employing this dual innovation outperform those utilizing conventional random masking, resulting in improved predictions of drug-related properties on ADMET benchmarks. This work contributes to the ongoing advancement of transformer architectures in the field of molecular representation learning.

## 1 Introduction

In machine learning, molecular graphs are frequently represented using SMILES strings, which translate molecular structures into sequences of tokens. Subsequently, transformers, powerful neural networks originally designed for natural language processing, have been adapted to learn molecular representations from SMILES by predicting randomly masked tokens within these sequences (Ross et al., 2022).

However, despite their success in other fields, transformers have under-presented in the leaderboards on the 22 ADMET benchmark datasets[1], which assess predictions of key drug properties—such as absorption, distribution, metabolism, excretion, and toxicity—that are crucial for determining the viability of drug candidates. In contrast, simpler fingerprint-based methods, which reduce molecular graphs into compact vectors summarizing the present of chemical motifs or functional groups, have achieved better results. For examples, Figure 1 shows a leaderboard of the top methods for toxicity prediction in the ADMET LD50_Zhu benchmark dataset where fingerprint-based and descriptor-based methods dominate.

The success of fingerprint-based methods highlights that understanding substructures, chemical motifs, or functional groups and their associations is crucial for predicting ADMET properties. This suggests that for transformers to be competitive, they need to enhance their ability to capture and interpret key molecular substructures and their relationships.

One possible reason for this performance gap is that transformers, when learning from SMILES strings, mask and predict tokens without considering the molecular substructures those tokens represent. Additionally, since the molecular graph is serialized into a string, transformers must recon-

---

[1] https://tdcommons.ai/benchmark/admet_group/overview/

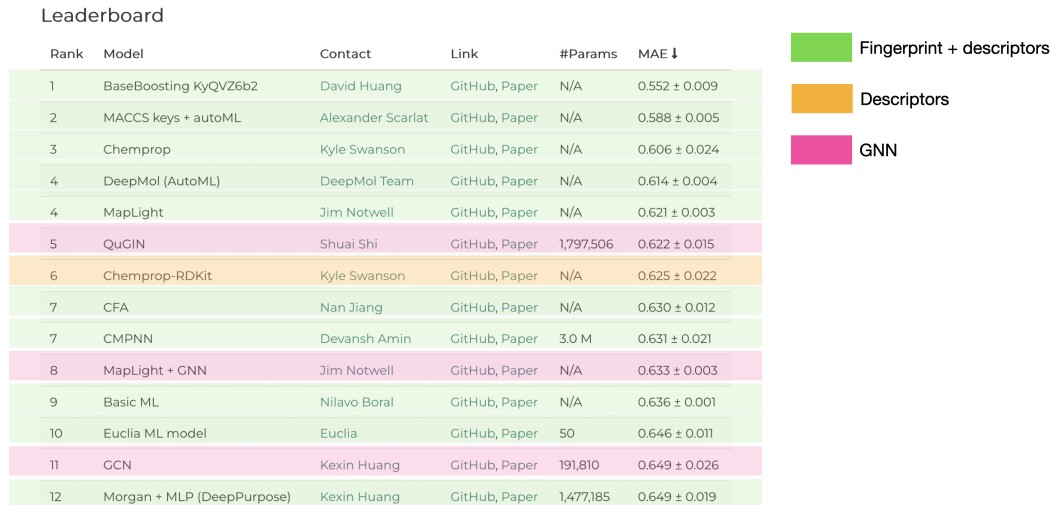

Figure 1: Leaderboard of top methods for toxicity prediction in the ADMET LD50_Zhu benchmark dataset. Fingerprint and descriptors based methods dominate.

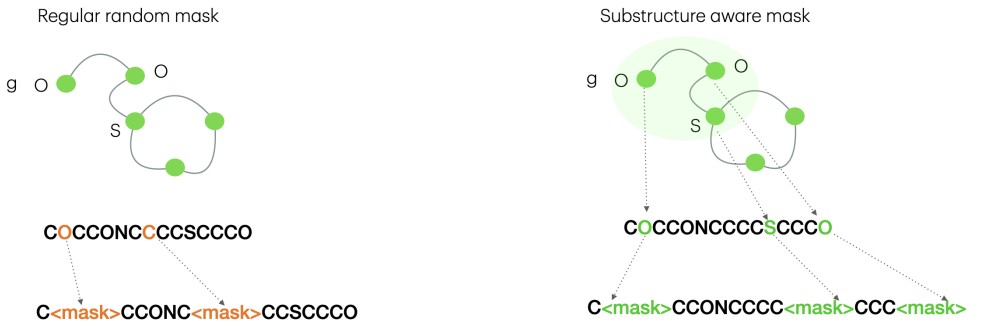

Figure 2: Random token masking (left) and substructure-aware masking (right).

struct these substructures from the string. Without an inductive bias to guide this string-to-graph translation, transformers require large amounts of data to learn these complex rules, making the task particularly challenging.

For instance, as shown in Figure 2 (left), random token masking in a SMILES string prompts transformers to rely on nearby tokens to predict the masked ones. This approach leads to transformers focusing on correlations between individual tokens—corresponding to atoms or bonds—without learning the more critical molecular substructures. While one might think that using a more advanced tokenizer, capable of grouping SMILES tokens into continuous substrings, could solve this issue, Figure 2 (right) illustrates that subgraphs like $g$ are not always represented by contiguous tokens in SMILES strings thus tokenization SMILES based on continuous substrings does not provide a solution for this issue.

To address this challenge, we introduce a novel method aimed at enhancing transformers' ability to recognize molecular substructures. The key idea is to incorporate substructure information directly into the masking and the prediction process, embedding an inductive bias that guides the model to more effectively learn molecular substructures. Specifically, our approach introduces a new masking strategy and a novel learning objective that leverage the molecular graph itself. Rather than masking random tokens, our model is trained to predict the presence of specific subgraphs. Groups of atoms, such as two oxygen atoms and one sulfur atom—representing molecular substructure $g$, as illustrated

in Figure 2 are masked together. This substructure-aware masking and the modified predicting objective enables the model to capture relationships within these substructures.

Our experiments demonstrate that transformers trained with the substructure-aware masking strategy outperform traditional random masking on 22 ADMET benchmarks, leading to more accurate predictions of drug-related properties. The proposed method ranked second in Elo rating among 34 baseline methods and achieved new state-of-the-art results in four tasks, highlighting its competitive advantage in improving transformer performance for molecular representation learning.

## 2 RELATED WORK

In recent years, learning molecular representation gained significant traction in molecular science, offering generalized frameworks for molecular tasks such as property prediction, drug discovery, and molecular generation. These models leverage large datasets to learn rich, transferable representations that can be fine-tuned for specific molecular applications. In this section, we discuss existing methods starting from classical fingerprint based methods to complex foundational molecular representation models.

### 2.1 FINGERPRINT BASED REPRESENTATION

Fingerprint-based and descriptor-based representations are popular techniques used in cheminformatics and computational biology for molecular characterization. A molecular fingerprint is a bit vector representation that captures the presence or absence of specific structural features in a molecule. Each bit corresponds to a particular substructure, functional group, or atom environment, encoding whether that feature exists in the molecule. Descriptors are quantitative properties that describe various attributes of a molecule. These include topological, geometric, electronic, and physicochemical properties. Unlike binary fingerprints, descriptors are typically numerical values that capture diverse molecular characteristics.

For the ADMET benchmarks, fingerprint and descriptor-based methods perform surprisingly well. Some of the top-performed methods such as MapLight (Notwell & Wood, 2023) which uses combination of MorganFingerprint (Rogers & Hahn, 2010), Avalon (Gedeck et al., 2006), ErG (Stiefl et al., 2006) and 200 descriptors from RDKit (Landrum, 2013). DeepMol (Correia et al., 2024) uses MorganFingerprints, MACCS Keys (Durant et al., 2002), and AtomPair Fingerprints (Map4 (Orsi & Reymond, 2024)). Among methods using fingerprints and descriptors we can also mention ChemProp (Yang et al., 2019), CFA (Quazi et al., 2023), ZairaChem (Turon et al., 2023). The proposed methods in the category are the established strong baselines that we will benchmark against in this work.

### 2.2 GRAPH NEURAL NETWORK

Graph-based models, particularly Graph Neural Networks (GNNs), are a natural fit for molecular data, where molecules are represented as graphs with atoms as nodes and bonds as edges. Early models such as Graph Convolutional Networks (GCNs) and Message Passing Neural Networks (MPNNs) have been widely applied to predict molecular properties. Gilmer et al. (2017) introduced MPNNs, which utilize message-passing mechanisms to capture the structure-property relationships within molecules, proving effective in tasks like quantum property prediction . Recent advancements, such as the integration of attention mechanisms GATs (Veličković et al., 2017), have improved the ability of these models to capture complex molecular interactions. The methods in this category are also present in the ADMET leaderboards even they are not as competitive as fingerprint based approaches.

### 2.3 TRANSFORMER BASED MODELS

Transformers, originally developed for natural language processing, have proven to be highly effective in molecular modeling, particularly for sequence-based tasks using SMILES strings. Models like ChemBERTa (Chithrananda et al., 2020) and MolFormer (Ross et al., 2022) leverage the transformer architecture to predict chemical properties from SMILES sequences with great success.

Meanwhile, Graphormer (Shi et al., 2022) extends transformers to graph-based molecular representations, combining self-attention mechanisms with inductive biases that capture both local and global molecular features, leading to state-of-the-art results in quantum property prediction. To ensure a fair and rigorous comparison, we implement and pretrain different transformer architectures on the same datasets as our proposed approach, controlling for dataset size variations that can otherwise skew performance outcomes. By doing so, we provide a more accurate assessment of each model's true capability in molecular property prediction.

### 2.4 MULTIMODAL MOLECULAR REPRESENTATION

Recent research investigates multimodal models that integrate molecular data with other modalities, such as 3D structural data, protein-ligand interactions, and experimental results. Foundational models like Uni-Mol, which rely on 3D information, or ImageMol, which utilizes molecular images, exemplify this approach. These models typically provide complementary information to transformer-based methods trained on SMILES representations. Since our work primarily focuses on transformers using SMILES data, we consider it orthogonal to methods trained on different modalities.

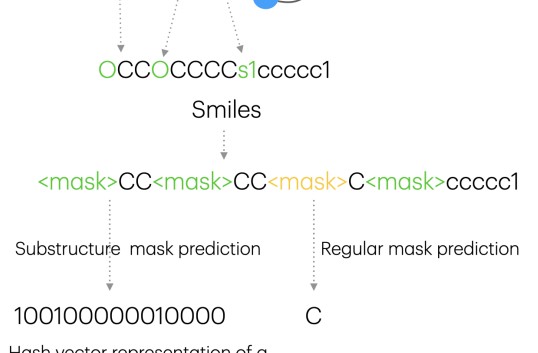

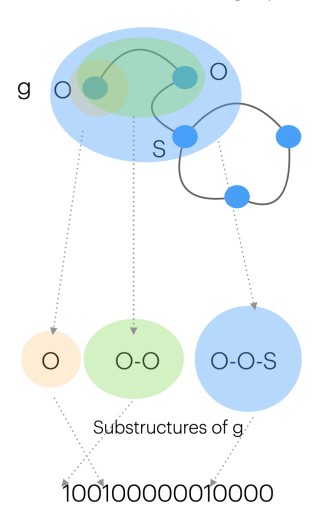

Figure 3: Left: an example of substructure masking (green) and individual token masking (orange). Right: construction of a substructure hash vector from a masked substructure.

## 3 ALGORITHM

In this section, we present an algorithm called SmilesGraph, developed to train transformers on recognizing high-order substructure relationships. The algorithm utilizes two forms of token masking, with the first being substructure masking. This method follows these steps:

- Randomly select an atom from the input molecular graph.
- Conduct a breadth-first traversal starting from the selected atom, continuing until reaching a maximum distance $r$ from the starting point.
- Record all atoms encountered during the traversal and map them to their positions in the SMILES string.
- Replace the identified atoms in the SMILES string with masking tokens.

An example of substructure masking is shown in Figure 3, where a substructure $g$ is derived by performing a breadth-first traversal from a starting node corresponding to an Oxygen atom, with a

Table 1: Hyperparameter settings

| Hyperparameters | Value |
|---|---|
| **Pretraining** | |
| Hash size | 256 |
| Substructure breadth-first traversal radius | 2 |
| The number of substructure masks | 1 |
| The number of regular token masks | 3 |
| Pretraining epoch | 1 |
| Pretraining learning rate max | 1e-5 |
| Pretraining learning rate min | 1e-6 |
| **Downstream** | |
| Downstream learning rate search range | 1e-5 to 1e-4 |
| The number of downstream FC layers search range | 1-4 |
| Downstream batch size search range | 2 to 256 |
| Downstream number of runs with different seeds | 5 |
| The number of folds in cross-validation | 5 |
| Early termination tolerance | 3 |

maximum traversal distance $r = 2$. In this example, three nodes—two corresponding to Oxygen atoms and one to a Sulfur atom—are masked within the associated SMILES string.

Besides substructure masking we also perform regular individual token masking (orange masks) so ensure that the model also learn regular token association.

Since a masked substructure must be represented as an entry in the alphabet, the number of possible masked tokens or substructures increases exponentially with the size of the substructure. To control the size of the alphabet, we use random hashing vectors to represent the alphabet. Specifically, for a given substructure, we construct the hash vector as follows:

- Retrieve the atom name of the starting node and compute its hash value within a hash vector of size $h$, setting the corresponding hash bit from 0 to 1.

- Expanding the starting node with its nearest neighbor nodes, concatenating the bonds and atom names of the neighboring nodes into a string. Compute the hash value of this string within a hash vector of size $h$, and set the corresponding hash bit from 0 to 1.

- Continue expanding the current nodes by including their neighbors, and repeat the process above.

Figure 3 (right) shows an example of how the hash vector of the subgraph $g$ was constructed. Starting from the oxygen node, it constructs string representation of the neighborhood of the node which results in O, O-O and O-O-S and the corresponding hash function is used to maps those strings into a hash vector with the corresponding bit is turns from zero to one. This method is similar to the Morgan fingerprint construction (Rogers & Hahn, 2010).

Once all the hash vectors are constructed, transformers are trained to predict the corresponding hash vectors. Since the hash vectors are fixed-size binary vector, we use a binary classifier head with BCE Loss (Binary Cross-Entropy Loss) as the training objective.

## 4 EXPERIMENTS

### 4.1 EXPERIMENT SETTINGS

**Training dataset** We used a subset of 100 million SMILES from the ZINC database (Irwin & Shoichet, 2005) to train our models, with 50,000 SMILES reserved for validation. All methods showed convergence with negligible differences when additional data was included. The dataset is available on our project website[2].

---

[2]anonymized GitHub repository, code in supplementary materials

Table 2: ADMET (Huang et al.) benchmark datasets statistics.

| Datasets | Task | Size | Split | Metrics |
|---|---|---|---|---|
| LD50_Zhu | Regression | 7385 | Scaffold | MAE |
| Caco2_Wang | Regression | 906 | Scaffold | MAE |
| Lipophilicity_AstraZeneca | Regression | 4200 | Scaffold | MAE |
| Solubility_AqSolDB | Regression | 9982 | Scaffold | MAE |
| PPBR_AZ | Regression | 1797 | Scaffold | MAE |
| Bioavailability_Ma | Classification | 640 | Scaffold | AUROC |
| HIA_Hou | Classification | 578 | Scaffold | AUROC |
| Pgp_Broccatelli | Classification | 1212 | Scaffold | AUROC |
| Substrate_CarbonMangels | Classification | 667 | Scaffold | AUROC |
| BBB_Martins | Classification | 1975 | Scaffold | AUROC |
| hERG | Classification | 648 | Scaffold | AUROC |
| AMES | Classification | 7255 | Scaffold | AUROC |
| DILI | Classification | 475 | Scaffold | AUROC |
| VDss_Lombardo | Regression | 1130 | Scaffold | Spearman |
| Half_Life_Obach | Regression | 667 | Scaffold | Spearman |
| Clearance_Hepatocyte_AZ | Regression | 1020 | Scaffold | Spearman |
| Clearance_Microsome_AZ | Regression | 1102 | Scaffold | Spearman |
| CYP2C9_Veith | Binary | 12092 | Scaffold | AUPRC |
| CYP2D6_Veith | Binary | 13130 | Scaffold | AUPRC |
| CYP3A4_Veith | Binary | 12328 | Scaffold | AUPRC |
| CYP2C9_Substrate_CarbonMangels | Binary | 666 | Scaffold | AUPRC |
| CYP2D6_Substrate_CarbonMangels | Binary | 664 | Scaffold | AUPRC |

**Benchmark datasets** We utilized 22 ADMET datasets from the Therapeutic Data Commons (Huang et al.) to benchmark various methods. The ADMET collection serves as a benchmark for evaluating the properties of chemical compounds in drug discovery and development. ADMET represents Absorption, Distribution, Metabolism, Excretion, and Toxicity—key pharmacokinetic and toxicological factors essential for determining the viability of drug candidates. These datasets help predict how a compound behaves in the human body, including its absorption, tissue distribution, metabolism by enzymes, excretion, and potential toxicity risks. The ADMET datasets were selected thanks to their active leaderboards, which include robust and diverse 32 baselines spanning both fingerprint-based and graph-based methods carefully tuned using AutoML and robust boosting tree models.

**Baseline methods** We compared our results with 32 baseline methods from the ADMET dataset leaderboards. The key methods are summarized below:

- DeepMol (Correia et al., 2024) addresses key challenges in the field, such as selecting optimal algorithms, automating data preprocessing, and ensuring consistent model performance across datasets. It automates critical steps in the ML pipeline, quickly identifying the best data representations, preprocessing techniques, and model configurations for molecular property or activity prediction.

- MapLight (Notwell & Wood, 2023) focuses on improving the prediction of small molecule using random forests and support vector machines paired with fingerprints (ECFP, Avalon, ErG) and 200 molecular properties.

- MapLight + GNN (Notwell & Wood, 2023) Similar to Maplight with the ensemble of MapLight with a GNN

- ChemProp (Yang et al., 2019) The paper evaluates two main classes of models for molecular property prediction: those using neural networks with computed molecular fingerprints or expert-crafted descriptors, and graph convolutional neural networks (GCNs) that work with a molecule's graph structure.

- CFA (Quazi et al., 2023) The paper focuses on improving the prediction of ADMET (Absorption, Distribution, Metabolism, Excretion, and Toxicity) properties, which are critical in drug discovery and development. Existing computational models often struggle with generalizability and robustness. The authors propose using Combinatorial Fusion Analysis (CFA) to enhance the performance of ADMET models.

Table 3: Performance of SmilesGraph compared to state-of-the-art (SOTA) results on regression tasks, evaluated using MAE metrics (lower is better). Bold values indicate current or newly achieved SOTA results.

| Datasets/Methods | LD50_Zhu | Caco2_Wang | Lipophilicity | Solubility | PPBR |
|---|---|---|---|---|---|
| **Pretraining** | | | | | |
| Transformers | 0.646 | 0.343 | 0.489 | **0.748** | 8.46 |
| Morgangen | 0.668 | 0.341 | 0.634 | 0.98 | 7.2 |
| SmilesGraph (Our) | 0.608 | 0.294 | **0.454** | 7.53 | **7.0** |
| **Fingerprint and Descriptors** | | | | | |
| DeepMol | 0.614 | 0.297 | 0.656 | 0.775 | 7.99 |
| MapLight | 0.621 | **0.276** | 0.525 | 0.792 | 7.660 |
| CFA | 0.630 | 0.341 | 0.626 | 0.939 | 8.680 |
| ChemProp | 0.606 | 0.330 | 0.467 | 0.761 | 7.788 |
| MACCS Keys + AutoML | 0.588 | - | - | - | 8.288 |
| Euclidia ML | 0.646 | 0.341 | 0.621 | 1.076 | 9.942 |
| DeepPurpose | 0.649 | 0.908 | 0.701 | 1.203 | 9.994 |
| BaseBoosting KyQVZ6b2 | **0.552** | - | 0.479 | - | 7.91 |
| Innoplexus ADME | - | - | 0.499 | 0.771 | 8.582 |
| **GNN** | | | | | |
| QuGIN | 0.622 | - | - | - | - |
| MapLight+GNN | 0.633 | 0.287 | 0.539 | 0.789 | 7.526 |
| GCN | 0.649 | 0.599 | 0.541 | 0.907 | 10.194 |

Table 4: Performance of SmilesGraph compared to state-of-the-art (SOTA) results on regression tasks, evaluated using Spearman correlation (higher is better). Bold values indicate current or newly achieved SOTA results.

| Datasets/Methods | VDss Lombardo | Half_Life Obach | Hepatocyte_AZ | Microsome_AZ |
|---|---|---|---|---|
| **Pretraining** | | | | |
| Transformers | **0.724** | 0.440 | 0.354 | 0.554 |
| Morgangen | 0.710 | 0.500 | 0.340 | 0.560 |
| SmilesGraph (Our) | 0.705 | 0.501 | 0.398 | 0.573 |
| **Fingerprint and Descriptors** | | | | |
| DeepMol | 0.497 | 0.485 | 0.440 | - |
| MapLight | 0.707 | 0.562 | 0.466 | 0.626 |
| CFA | 0.628 | **0.576** | 0.536 | 0.625 |
| ChemProp | 0.491 | 0.265 | 0.431 | 0.599 |
| Euclidia ML | 0.609 | 0.547 | 0.424 | 0.572 |
| DeepPurpose | 0.561 | 0.329 | 0.382 | 0.586 |
| Voting Regressor | - | 0.544 | - | - |
| RFStacker | - | - | - | 0.625 |
| Innoplexus ADME | 0.707 | 0.511 | 0.457 | - |
| **GNN** | | | | |
| MapLight+GNN | 0.713 | 0.557 | **0.498** | **0.630** |
| GCN | 0.457 | 0.239 | 0.366 | 0.532 |
| SimGCN | 0.582 | 0.392 | - | 0.597 |

- ZairaChem (Turon et al., 2023) an AI- and ML-based tool designed for small-molecule activity prediction. ZairaChem is fully automated, requiring minimal computational resources, and can handle a wide range of datasets, including assays for cell growth inhibition and drug metabolism properties.
- SimGCN[3] a method based on GCN network.
- CNN (DeepPurpose) (Huang et al., 2020) a method based on Convolution Neural Network.
- RDKit2D + MLP (DeepPurpose) (Huang et al., 2020) descriptors and MLP

Besides we also compare our approach to the following pretrained methods that are not part of the leaderboards:

- Transformers: we train a language model with the same backbone as SmilesGraph and pretrained the model with the same training dataset. This model is used to validate the proposed learning methods in SmilesGraph.
- MorganGen Hoang et al. (2024): is a generative model that is trained on the same training set that learns to generate SMILES from MorganFingerprint.

**Downstream models** For each of the methods—SmilesGraph, Transformers, and MorganGen—we incorporate classification or regression heads with alternative fully connected layers and ReLU activation functions on top of the transformers. Given the small size of the ADMET datasets, we use 5-fold cross-validation on the training sets, repeating the process with 5 different random seeds. Hyperparameter optimization (HPO) was performed to identify the best hyperparameters that maximize cross-validation performance on the training data. Table 1 summarizes the HPO search space. The final predictions are based on the average of the models trained across the folds, and these predictions are used to report the test set results.

**Code and Model Availability** We have included the code as supplementary material for this paper. The source code and pretrained models will be published on GitHub and HuggingFace, with the links anonymized to maintain double-blind review.

**Hyperparameters and Experimental Settings** Table 1 outlines the hyperparameter settings employed in our experiments. A detailed sensitivity analysis of key hyperparameters is presented in the subsequent ablation study. In terms of computational resources, we utilized two NVIDIA A100 GPUs. Pretraining each model for one epoch took approximately three weeks.

### 4.2 RESULTS ON ADMET AND DISCUSSION

Tables 3-6 present a comparative study of SmilesGraph against top-performed baseline approaches from the current ADMET leaderboards. The following conclusions can be drawn from the results:

- SmilesGraph, incorporating the new masking methods and learning objectives, outperformed transformer-based methods using the same pretraining data and downstream experimental settings. Specifically, it either outperformed or was competitive with transformers in 7 out of 9 regression tasks and in 16 out of 22 tasks overall. The significant performance of SmilesGraph on regression tasks suggests that learning substructure associations is crucial for better results.
- Compared to the SOTA results (the best among baseline algorithms), SmilesGraph achieved new SOTA performance in 4 tasks. Notably, it significantly reduced the prediction error from 7.5 to 7.0 in the PPBR prediction task, representing a 6.6% improvement, and reduced the error from 0.525 to 0.454, equivalent to a 13.5% improvement.

The results in Tables 3-6 indicate that no single method consistently outperforms the others across all ADMET datasets. To provide a more holistic comparison, we utilize Elo ratings [4], assigning each method an initial score of 1500. Pairwise comparisons are conducted to update the Elo scores based on the relative performance of each method. Figure 4 (left) shows the Elo ratings of SMILES

---

[3]`https://github.com/KatanaGraph/SimGCN-TDC/blob/main/Report_SimGCN_for_`
`TDC_Benchmarks.pdf`

[4]`https://en.wikipedia.org/wiki/Elo_rating_system`

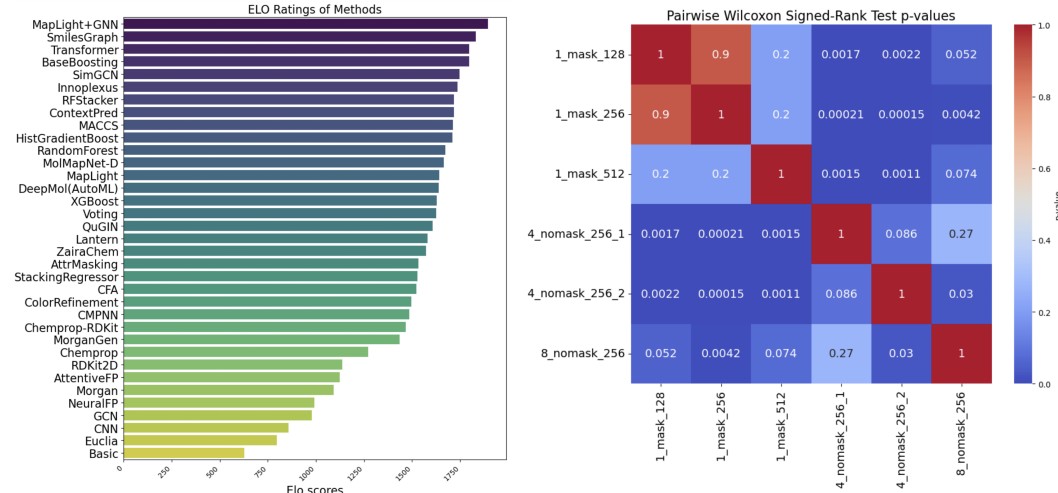

Figure 4: Elo ratings (left) and Wilcoxon Signed rank test (right) in an ablation study. Overall SmilesGraph ranked 2nd in terms of Elo rating.

Table 5: Performance of SmilesGraph compared to state-of-the-art (SOTA) results on classification tasks, evaluated using AUROC (higher is better). Bold values indicate current or newly achieved SOTA results.

| Data/Methods | Bioavail. | HIA | Pgp | CYP3A4 Substrate | BBB Martins | hERG | AMES | DILI |
|---|---|---|---|---|---|---|---|---|
| **Pretraining** | | | | | | | | |
| Transformers | 0.704 | 0.984 | 0.917 | 0.650 | 0.907 | 0.831 | 0.870 | 0.936 |
| Morgangen | 0.65 | 0.97 | 0.92 | 0.59 | 0.84 | 0.74 | **0.89** | 0.86 |
| SmilesGraph | 0.718 | 0.985 | 0.926 | 0.660 | 0.907 | 0.845 | 0.866 | **0.939** |
| **Fingerprint and Descriptors** | | | | | | | | |
| DeepMol | **0.753** | **0.990** | 0.922 | 0.655 | 0.868 | 0.763 | 0.847 | 0.885 |
| MapLight | 0.730 | 0.986 | 0.930 | 0.650 | 0.916 | 0.871 | 0.868 | 0.887 |
| CFA | 0.746 | 0.981 | 0.928 | **0.667** | **0.920** | 0.875 | 0.852 | 0.919 |
| ChemProp | 0.606 | 0.981 | 0.886 | 0.629 | 0.869 | 0.840 | 0.850 | 0.899 |
| Euclidia ML | 0.613 | 0.926 | 0.845 | 0.629 | 0.725 | 0.749 | 0.755 | 0.873 |
| DeepPurpose | 0.672 | 0.972 | 0.918 | 0.662 | 0.889 | 0.841 | 0.823 | 0.875 |
| ZairaChem | 0.706 | 0.948 | 0.935 | 0.630 | 0.910 | 0.656 | 0.871 | 0.925 |
| **GNN** | | | | | | | | |
| MapLight+GNN | 0.742 | 0.989 | **0.938** | 0.647 | 0.913 | **0.880** | 0.869 | 0.917 |
| GCN | 0.566 | 0.936 | 0.895 | 0.590 | 0.842 | 0.738 | 0.818 | 0.859 |
| SimGCN | 0.748 | - | 0.929 | 0.640 | 0.901 | 0.874 | - | 0.904 |

in comparison to 34 baseline methods. SmilesGraph achieved the second-highest rating, surpassed only by the MapLight+GNN ensemble. Since MapLight+GNN combines fingerprint, descriptor, and graph-based techniques, future work could explore whether integrating transformers with these approaches might further improve predictive performance.

## 4.3 ABLATION STUDY

In this section, we conduct experiments to evaluate the impact of different components, including substructure hash size, the number of masked substructures, and the comparison between masking and non-masking of substructure tokens.

Due to the time-intensive nature of training—each model takes 4 weeks on our available GPU resources—we did not explore all possible combinations of these factors. Instead, we varied the hash size across $\{128, 256, 512\}$ and the number of masked substructures across $\{1, 4, 8\}$. Additionally,

Table 6: Performance of SmilesGraph compared to state-of-the-art (SOTA) results on binary classification tasks, evaluated using AUPRC (higher is better). Bold values indicate current or newly achieved SOTA results.

| Datasets/Methods | CYP2C9 Veith | CYP2D6 Veith | CYP3A4 Veith | CYP2C9 Substrate | CYP2D6 Substrate |
|---|---|---|---|---|---|
| **Pretraining** | | | | | |
| Transformers | 0.804 | 0.701 | 0.888 | 0.422 | 0.735 |
| Morgangen | 0.76 | 0.650 | 0.86 | 0.40 | 0.70 |
| SmilesGraph (Our) | 0.798 | 0.698 | 0.887 | 0.422 | **0.738** |
| **Fingerprint and Descriptors** | | | | | |
| DeepMol | 0.758 | 0.685 | - | 0.417 | 0.731 |
| MapLight | 0.783 | 0.723 | 0.881 | 0.415 | 0.713 |
| CFA | 0.751 | 0.664 | 0.855 | 0.417 | 0.704 |
| ChemProp | 0.777 | 0.673 | 0.876 | 0.400 | 0.686 |
| Euclidia ML | 0.536 | 0.348 | 0.639 | 0.347 | 0.498 |
| DeepPurpose | 0.742 | 0.616 | 0.829 | 0.380 | 0.677 |
| ZairaChem | 0.786 | 0.644 | 0.875 | **0.441** | 0.685 |
| **GNN** | | | | | |
| MapLight+GNN | **0.859** | **0.790** | **0.916** | 0.437 | 0.720 |
| GCN | 0.735 | 0.616 | 0.840 | 0.344 | 0.617 |
| SimGCN | - | - | - | 0.433 | - |
| ContextPred | 0.839 | 0.739 | 0.904 | 0.392 | 0.736 |

we examined the effect of masking versus non-masking of substructure tokens while keeping other factors constant.

Figure 4 shows p-values from the Wilcoxon signed-rank test comparing the SmilesGraph model across different settings: substructure count, masking, and hash vector size (e.g., 8_nomask_256 indicates a model with no masking, a hash size of 256, and 8 substructures). The results demonstrate that models using substructure masking significantly outperform those without. While the hash size does not have a statistically significant effect, smaller hash sizes tend to yield slightly better results. Additionally, the number of masked substructures has minimal impact in the no-mask setting.

## 5 CONCLUSION AND FUTURE WORK

In this paper, we introduced a substructure-aware masking strategy coupled with a novel learning objective to enhance transformer-based models for molecular representation learning using SMILES. Our approach addresses the limitations of traditional random token masking, which fails to capture essential molecular substructures and often leads to suboptimal performance on crucial ADMET tasks. By embedding molecular substructure information directly into the masking and prediction process, our method enables transformers to better model the relationships within these substructures, improving their ability to predict drug-related properties. Empirical evaluations demonstrate that our approach significantly outperforms conventional random masking techniques on ADMET benchmarks, highlighting the importance of incorporating domain-specific molecular knowledge into the design of transformer models.

While our proposed method has shown promising results, several avenues for future research remain. First, further refinement of the substructure-aware masking strategy could be explored by incorporating more sophisticated chemical knowledge, such as reaction mechanisms or quantum properties, to improve model generalization across diverse molecular datasets. Additionally, applying the approach to other molecular representation formats beyond SMILES, such as molecular graphs or 3D structures, could broaden its applicability.

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
