# OpenReview forum: "Learning High-Order Substructure Association from Molecules with Transformers"
_ICLR.cc/2025/Conference — ICLR 2025 Conference Withdrawn Submission_

### Official Review · Reviewer_udzS · 2024-10-16

**Soundness:** 2
**Presentation:** 2
**Contribution:** 1
**Rating:** 3
**Confidence:** 4

**Summary:**

The authors present a novel approach aims at improving transformers' ability to identify molecular substructures through a substructure-aware masking strategy. They utilize molecular graphs to identify groups of neighboring molecules based on a radius r (referred to as substructures) and map the corresponding characters in SMILES strings. Subsequently, they apply a hash function to the identified substructures and use a masked language modeling transformer to predict fixed-size binary hash vectors, employing binary cross-entropy loss for optimization.

**Strengths:**

- The authors thoroughly evaluate their approach across a wide range of tasks (22 ADMET tasks).
- The method achieves state-of-the-art results on 4 out of 22 tasks, highlighting its effectiveness and potential in specific applications within the field.
- The authors conduct a comprehensive ablation study examining the impact of hash size h and the number of masked substructures. This rigorous analysis provides valuable insights into the model's performance and helps identify optimal configurations for enhancing predictive accuracy.

**Weaknesses:**

Methodology

Substructure-aware masking strategy:
- The authors randomly choose an atom and expand it to a specific radius r to form the substructure for masking. However, this method does not effectively capture chemically meaningful substructures, such as functional groups like OH, COOH, and others. By relying on a purely random selection of atoms and a fixed radius, the approach may overlook essential chemical features that are important for accurately modeling molecular behavior. Instead, a more effective strategy would be to focus on well-defined functional groups or significant chemical motifs. This could enhance the representation of molecules and improve the understanding of molecular substructure interactions and properties.

Hash Vector Generation:
- The explanation lacks sufficient detail regarding the characteristics and implementation of the hash vector. It would be beneficial for the authors to clarify how the hash function is defined and how collisions are managed. Without this information, the effectiveness of the hashing process remains uncertain.
- Random hashing can lead to information loss, particularly if multiple unique substructures hash to the same value (collisions). The authors should address how they minimize the loss of information due to collisions and whether they evaluate the effectiveness of this method in retaining meaningful structural information.

Masked Language Modeling:
- In line 237, the rationale for performing regular individual token masking is unclear. The authors state that it is "to learn regular token association," but the specific purpose of this masking remains ambiguous.
- The authors should conduct an ablation study using the transformer model under the same experimental conditions, differing only in the masking strategy. This would provide a clearer demonstration of the effectiveness of the proposed substructure-aware masking strategy compared to conventional approaches.

Evaluation:
The authors benchmark their method across a wide range of ADMET tasks, reporting results on 4 out of 22 tasks. However, I recommend that the authors also run benchmarks on MoleculeNet tasks to compare their performance against other well-established molecular representation methods.

Presentation:
- The first sentence of the paper states, "Molecular graphs are commonly represented using SMILES (Simplified Molecular Input Line Entry System) strings," which is not entirely accurate. Molecular graphs and SMILES represent molecules in different ways; therefore, it is misleading to assert that SMILES represent molecular graphs.
- In the abstract, the authors claim to propose "a new learning objective," but it is unclear what this new learning objective entails. If it refers to the BCE loss, then BCE loss is not a new concept.
- In line 239, the term "alphabet" in the context of masked substructures could be misleading, as "alphabet" typically refers to a fixed set of symbols. It would be clearer to refer to it as a "vocabulary" or "set of possible tokens," aligning the terminology with common practices in natural language processing (NLP).
- The figures depicting molecular graphs in the paper do not accurately represent the structures of the molecular graphs discussed.
- Citations for the baseline models in Tables 3, 4, and 5 should be included for proper attribution and context.
- The writing in the paper is not well polished, and there are several grammatical errors that need to be addressed to improve clarity and comprehension.

**Questions:**

- Clarification of the New Learning Objective: Could you clarify what you mean by "a new learning objective"? Specifically, how does this differ from established methods like binary cross-entropy (BCE) loss?

- Impact of Masked Substructures: You mentioned that "the number of masked substructures has minimal impact in the no-mask setting." Can you elaborate on this finding? What implications does this have for the design of your substructure-aware masking strategy?

---

### Official Review · Reviewer_hPpg · 2024-10-31

**Soundness:** 2
**Presentation:** 1
**Contribution:** 1
**Rating:** 3
**Confidence:** 3

**Summary:**

This paper proposes the use of substructure masking, instead of random masking, for molecular SMILES representation learning and achieves performance gains on ADMET tasks.

**Strengths:**

The paper is well-structured and covers most of the important sections.

**Weaknesses:**

1.  **Novelty**: The proposed approach seems overly simple and lacks novelty, especially given that substructure masking for graph or SMILES has been extensively explored in recent work (e.g., GROVER[1], UniMAP[2], Unicorn[3]). This paper’s masking approach could be seen as a subset of UniMAP's method, but lacks a clear advantage or innovation in comparison. Specifically, UniMAP’s Fragment Level Cross-Modality Masking task masks the substructure of the graph and its corresponding smiles sequence, which is almost the same as the idea of ​​this paper.

[1] Self-Supervised Graph Transformer on Large-Scale Molecular Data, NeurIPS 2020

[2] UniMAP: Universal SMILES-Graph Representation Learning, 2023

[3] UniCorn: A Unified Contrastive Learning Approach for Multi-view Molecular Representation Learning, ICML 2024

2. **Writing and Literature Support**:

The motivation in the introduction requires stronger literature support. For example, the claim that transformer-based models are inferior to description/fingerprint models would benefit from additional evidence and analysis.

The Related Work section lacks references in subsection 2.4. The methodology section presents only the masking algorithm without introducing the overall model architecture, specific loss functions, or details like masking ratios.

Reference formatting is inconsistent: for instance, the citation for the ADMET benchmark (Huang et al.) is missing a publication year, and the reference for Graphormer should include its original proposal, not solely later benchmarking studies.

3. **Experimental Evaluation**:

The experimental evaluation is limited, as it only uses one benchmark and lacks sufficient transformer-based baselines. The reported results do not convincingly demonstrate the effectiveness of the approach. In most tasks, the proposed method does not surpass baselines.

Recommended baselines to consider include transformer-based methods like ChemBERTa and MolFormer. Given the convertibility between SMILES and 2D graphs, including graph masking methods as baselines (such as Graphormer and GROVER) would add relevance.

The experimental setting lacks detail, particularly on Elo ratings and Wilcoxon signed-rank test definitions.

---
Based on the issues outlined above, I find the method overly simplistic and lacking in noticing recent works. The writing requires substantial improvements, and the experimental improvements do not appear sufficient for acceptance at a top conference. Therefore, I recommend rejection.

**Questions:**

1. Could the authors incorporate more recent baselines and potentially test on the MolNet benchmark?

2. How does this method compare to other substructure masking approaches like UniMAP and GROVER in terms of advantages or improvements?

3. For the Wilcoxon signed-rank test, was a one-tailed or two-tailed test applied? Please clarify the specific alternative hypotheses in each pairwise comparison in paper.

4. For the leaderboard in Figure 1, could the authors specify the time of the ranking data collection?

---

### Official Review · Reviewer_8aHp · 2024-10-31

**Soundness:** 2
**Presentation:** 2
**Contribution:** 1
**Rating:** 3
**Confidence:** 4

**Summary:**

The paper introduces a substructure-aware masking strategy alongside a new learning objective. Compared with random masking, the proposed masking strategy results in improved predictions.

**Strengths:**

1. The paper is well-motivated since structure-aware masking provides more domain knowledge.
2. The paper is evaluated on extensive experiments.

**Weaknesses:**

1. The improved masking strategy does not provide significant improvement in experiments. In regression tasks, since the overall performance of SmilesGraph does not outperform MapLight+GNN, does this means it can be better just ensemble two models instead of providing structural knowledge in masking strategy?

2. The baselines are not comprehensive enough. Many advanced Transformer-based achieved excellent performance, such as MolT5[1] and BioT5[2]. It will be much more convincing if there's direct comparison among these approaches.

3. The major novelty, masking the molecule based on the substructure, is somewhat incremental.

[1]. Translation between Molecules and Natural Language
[2]. BioT5: Enriching Cross-modal Integration in Biology with Chemical Knowledge and Natural Language Associations

**Questions:**

See weaknesses.

---

### Official Review · Reviewer_BeZK · 2024-11-04

**Soundness:** 2
**Presentation:** 1
**Contribution:** 1
**Rating:** 1
**Confidence:** 4

**Summary:**

This paper introduces a new mask strategy for pretraining the smiles-based Transformers. In contrast to the previous token-level masked strategy, the proposed method tends to conduct a substure-level prediction task during pretraining to better utilize the inductive bias over the molecule data. The experiments are conducted over several property prediction benchmarks to demonstrate the effectiveness of the proposed method.

**Strengths:**

1. I believe that the paper is studying a reasonable problem of mask-predicted based objective. And integrating the new masked strategy could be an interesting exploration direction.
2. The paper makes a comprehensive review and discussion of the previous works, including the current leaderboard of the several important benchmarks. And it is helpful for gaining an intuitive overview of the progress in the field.

**Weaknesses:**

However, I believe the paper needs a major revision to get accepted by top conferences.
1. The presentation and formulation of the paper is very poor and it seems the paper is finished in a rush. Specifically, the paper is making some basic misunderstandings by treating Transformer as a training algorithm which is actually a network structure. And it causes confusion for me to understand the actual training algorithm. I suppose it is a mask prediction like Bert's objective. If the guess is correct, what are the details for the masking? e.g. What ratio of the tokens is masked during the pretraining?
2. The experiments results are not solid enough. The main results are conducted over several benchmarks. However, whether the training data of different methods are aligned is lack of explaination. Besides, the ablation study is also missed to demonstrate the effectiveness of different components.
3. Though I believe the motivation of the method is reasonable, the substructure or scaffold based tokenizers are proposed several years ahead of this work, e.g. JTVAE. I would like to know what the benefit of the proposed approach is over using fragment tokenizer and conducting masked predictions.
Given the above  facts, I recommend a strong rejection of this draft.

**Questions:**

Refer to above weakness.

---

### Note · Authors · 2024-11-21

I have read and agree with the venue's withdrawal policy on behalf of myself and my co-authors.